# Associations Between Overparenting and Offspring’s Mental Health: A Meta-Analysis of Multiple Moderators

**DOI:** 10.3390/bs15091235

**Published:** 2025-09-11

**Authors:** Na Hu, Kewan Chen, Longying Ye, Hongjin Liu, Dan Cai, Huafeng Zhang, Yanli Zhao

**Affiliations:** 1School of Psychology, Shanghai Normal University, Shanghai 200234, China; huna54@shnu.edu.cn (N.H.); chenkewan12@163.com (K.C.); y3287716508@126.com (L.Y.); 1000555026@smail.shnu.edu.cn (H.L.); caidan@shnu.edu.cn (D.C.); 2Lab for Educational Big Data and Policymaking, Ministry of Education of the People’s Republic of China, Shanghai 200034, China; 3Research Institute for International and Comparative Education, Shanghai Normal University, Shanghai 200234, China; 4Psychiatric Research Center, Beijing Huilongguan Hospital, Peking University Huilongguan Clinical Medical School, Beijing 100096, China

**Keywords:** mental health, overparenting, depression, anxiety, meta-analysis

## Abstract

Despite overparenting being considered a critical factor associated with offspring’s mental health, the existing research on this topic has yielded inconsistent results. The current study aimed to examine the relationship between overparenting and various mental health outcomes, such as anxiety, depression, life satisfaction, and subjective well-being, through an analysis of effect size. Furthermore, potential moderators, including developmental stage (indexed by age), gender, information informants, study design, and cultural factors, were also examined. A total of 44 studies (*N* = 21,607) were identified in the meta-analysis, with 34 studies examining anxiety, 32 studies examining depression, 13 studies examining life satisfaction, and three studies examining subjective well-being. The results revealed a positive yet small association between overparenting and mental health indicators (*r*_anxiety_ = 0.16, *r*_depression_ = 0.20, *r*_life satisfaction_ = 0.09, *ps* < 0.001), except for subjective well-being (*r*_subjective well-being_ = 0.09, *p* > 0.001). Moderator analyses showed that the high heterogeneity across studies was explained by culture, parents’ gender, and developmental stage. These findings emphasize that overparenting is linked to mental health issues, particularly anxiety and depression. This study also suggests that heterogeneity should be considered for future clinical interventions and parenting-based educational programs.

## 1. Introduction

Mental health disorders are a leading contributor to the global burden of disease, significantly impacting both mortality and daily functioning ([74]). As their prevalence continues to rise, research concerning the influencing factors of mental health disorders is also increasing ([70]). Among these, a contemporary form of parenting practices—commonly labeled overparenting—has emerged as a salient yet contested factor influencing offspring’s mental health. A clearer understanding of this relationship is essential for informing effective prevention and intervention strategies.

Empirical studies have primarily focused on the negative associations between overparenting and offspring’s mental health, yet some research has also identified positive or neutral links, resulting in inconsistent findings. These mixed results highlight the need for a more comprehensive framework to capture both the detrimental and beneficial aspects of mental health. From a theoretical perspective, the Dual-Factor Model of Mental Health emphasizes the importance of examining both psychopathological symptoms (e.g., depression, anxiety) and positive indicators (e.g., life satisfaction, subjective well-being) to gain a more comprehensive understanding of mental health outcomes.

To address these gaps, the present study conducts a meta-analysis examining the relationship between overparenting and four key indicators of mental health: anxiety, depression, life satisfaction, and subjective well-being. Drawing on a systematic review of existing literature, this meta-analysis also aims to generate evidence-based recommendations for family-based interventions targeting adolescent mental health. In doing so, the study offers a theoretical and empirical foundation for personalized clinical treatment strategies and more effective mental health promotion.

### 1.1. Conceptions of Overparenting

Overparenting, also known as helicopter parenting, refers to developmentally inappropriate parental behaviors that characterized by high levels of control, protection, and involvement that exceed age-normative expectations ([62]; [13]). These behaviors often manifest as anticipatory problem-solving, risk aversion, and the provision of unnecessary assistance, limiting children’s autonomy and hindering the development of self-efficacy and independent problem-solving skills ([39]; [12]). Importantly, overparenting is a relative concept, defined by behaviors that exceed what is considered appropriate for a child’s age or cultural norms. Typical overparenting behaviors include: (1) Making decisions or solving problems on behalf of the child, including intervening in daily affairs or shielding the child from negative outcomes ([31]; [39]). (2) Providing excessive advice or directives, limiting opportunities for independent choice ([39]; [12]). (3) Preventing manageable challenges, through risk aversion or unnecessary assistance, which reduces learning from experience ([39]; [12]). By contrast, normative parental involvement offers age-appropriate guidance and support that fosters autonomy and problem-solving. Unlike typical parenting, overparenting exceeds developmental norms, potentially impeding autonomy, self-efficacy, and a sense of control ([31]). The psychological impact of overparenting may vary: while it can potentially impede autonomy and self-efficacy, in some developmental stages or cultural contexts, it may also be perceived as warmth or support, highlighting the nuanced and context-dependent nature of this parenting behavior.

### 1.2. Overparenting and Offspring’s Mental Health

Empirical research has yielded mixed findings regarding the mental health outcomes associated with overparenting. Importantly, mental health encompasses both negative indicators, such as anxiety and depressive symptoms, and positive indicators, which reflect positive mental health, including life satisfaction and subjective well-being ([65]). In this study, positive indicators are used to operationalize positive mental health, providing measurable manifestations of the broader construct. Examining both types of outcomes provides a more comprehensive understanding of offspring’s psychological functioning and allows for the identification of potential “double-edged” effects of overparenting, which may simultaneously undermine negative outcomes while supporting positive aspects of well-being.

On the one hand, a substantial body of work, primarily in Western contexts, links overparenting to elevated anxiety, depressive symptoms, and emotional distress in adolescents and emerging adults ([36]; [59]; [53]; [11]; [32]). Theoretically, Self-Determination Theory ([56]) posits that the satisfaction of basic psychological needs—such as autonomy, competence, and relatedness—is essential for mental health ([57]). Overparenting may thwart autonomy by limiting children’s decision-making and problem-solving experiences. Similarly, Separation-Individuation Theory ([19]) suggests that excessive parental control disrupts the normative process through which adolescents establish independence from their parents, thereby increasing their vulnerability to psychological distress ([52]).

On the other hand, some studies, particularly those conducted in East Asian cultural contexts, suggest that parental overinvolvement may be associated with neutral or even positive mental health outcomes, such as lower depressive symptoms and greater life satisfaction ([35]; [33]). In these cultural contexts, parental overinvolvement may be interpreted as a form of familial support or interdependence rather than intrusion. [16] ([16]), for example, found that strong parent–child support under overparenting conditions was linked to improved well-being in adult children. These findings highlight the significance of cultural interpretations in influencing the psychological outcomes of specific parenting behaviors.

Despite the growing body of research on overparenting, existing reviews and meta-analyses have primarily focused on its negative psychological consequences ([75]) and have largely overlooked culturally diverse samples and positive indicators of mental health. Most notably, the inconsistent findings across cultural contexts and study design suggest that the strength and direction of associations may vary depending on specific conditions. These considerations highlight the need to examine potential moderators that could account for when and why overparenting is associated with either detrimental or beneficial mental health outcomes.

### 1.3. Potential Moderators of Associations Between Overparenting and Mental Health

As highlighted in the previous section, the associations between overparenting and offspring’s mental health appear to vary across studies and contexts. To better understand this variability, it is essential to examine potential moderating factors that may influence when and why overparenting leads to detrimental, neutral, or even beneficial outcomes.

First, the gender of the offspring and parents has been proposed as a potential moderator in this relationship. For instance, studies have shown that paternal overparenting has a positive correlation with adolescents’ life satisfaction. Maternal overparenting positively correlates with adolescents’ anxiety and depression ([37]). Furthermore, high levels of overparenting have been linked to lower levels of happiness in women ([32]). One study found that maternal helicopter parenting was associated with excessive interpersonal sensitivity in daughters and lower levels of interpersonal sensitivity in sons. In contrast, paternal helicopter parenting was associated with more negative outcomes, regardless of the child’s gender ([55]). However, it remains unclear whether differences in child or parent gender impact the relation between overparenting and positive aspects of mental health, particularly the effect of child gender differences on overparenting and life satisfaction.

Second, the distinction between the individualistic culture predominant in the West and the collectivist culture prevalent in the East may lead to varying associations between overparenting and mental health, particularly in East Asia, where overparenting could be linked to positive mental health ([35]). A prior cross-cultural investigation found that overparenting is correlated with reduced subjective well-being among American and Chinese students ([22]). Some researchers have examined the relationship between overparenting and academic achievement among American and Korean college students, finding cultural differences in their findings ([29], [30]). Therefore, further exploration is warranted to determine whether cultural background serves as a moderating factor in the relationship between overparenting and mental health.

Moreover, developmental stage, as indexed by offspring age, may significantly influence the relationship between overparenting and mental health. Leung pointed out that the age of children is one of the factors that influence overparenting ([37]). From a developmental perspective, children in early childhood typically require guidance and support to navigate challenges, and parental involvement may be experienced as supportive rather than intrusive. In contrast, adolescents and emerging adults increasingly seek autonomy, and overparenting may thwart their growing need for independence, leading to elevated anxiety, depressive symptoms, or reduced life satisfaction ([56]; [19]). Therefore, developmental stage could theoretically moderate the impact of overparenting on mental health outcomes. Empirically, most prior studies examining the associations between overparenting and mental health have focused on adults, particularly emerging adults ([9]; [52]) whereas research on children or adolescents remains limited ([62]; [10]). Consequently, it remains unclear whether developmental stage moderates the relationship between overparenting and mental health. This study primarily aims to explore potential differences in the effects of overparenting on children’s mental health across developmental stages.

Other moderating variables that warrant exploration in this study include the information provider and the study design. Particularly, study design may influence observed associations. Most prior studies are cross-sectional, capturing associations at a single time point, which limits the ability to infer causality ([68]). In contrast, longitudinal or experimental designs can more accurately examine the temporal and causal relationships between overparenting and offspring mental health, and may yield different effect sizes or moderation patterns. Thus, study design is considered a potential moderator in the current meta-analysis.

### 1.4. The Present Study

The present meta-analysis systematically examines the relationship between overparenting and both negative (depression, anxiety) and positive (subjective well-being, life satisfaction) mental health outcomes. It also examines potential moderators, including offspring and parent gender, developmental stages, cultural background, and study design. Given the existing evidence and aims of the current study, we hypothesized that overparenting is linked with indicators of mental health, varied by the moderators mentioned. By integrating findings across diverse cultural contexts and methodological variations, this study aims to provide a more nuanced understanding of the impact of overparenting on offspring’s mental health and inform future intervention and prevention efforts.

## 2. Method

### 2.1. Search Procedure and Selection Steps

*Search procedure*. We identified relevant journal articles and academic papers across various electronic databases, including PsycINFO, PsycArticles, Scopus, Google Scholar, Web of Science, ProQuest, and Chinese databases such as CNKI and WanFang.

Studies published from 2003 to October 2023 were included in the literature search using Google Scholar, which included the top 100 articles deemed most relevant according to previous studies ([75]). We used specific search terms as follows to search the title and abstract of the article: (overparenting* OR over-parenting* OR helicopter parenting* OR overinvolvement* OR overhelp* OR hyper-parenting* OR overprotect parenting* OR “intent* parent* OR hover* OR tiger parent* OR overcontrol* OR excessive parent* OR excessive involve* OR excessive assist* OR excessive help*) AND (mental health * OR depression * OR depression symptoms* OR anxiety* OR internalizing problems* OR life satisfaction* OR subjective well-being*).

*Inclusion criteria*. Articles were included if they: (a) assessed parental overparenting or helicopter parenting; (b) assessed mental health, including depressive symptoms, anxiety, life satisfaction, and subjective well-being; (c) full texts were sourced from peer-reviewed English journals and Chinese journals; (d) included participants without clinically diagnosed psychiatric or neurological disorders, to avoid confounding the associations between overparenting and mental health outcomes. Minor or transient physical conditions (e.g., common colds) were not grounds for exclusion; (e) results indicate effect size indicators.

*Selection steps*. A PRISMA flow diagram (see Figure 1) illustrates the study selection steps. A total of 3779 articles were identified through database searches. After removing 1171 duplicates, 2608 articles were screened by title and abstract, which was conducted by two coders. Of these, 2382 articles were excluded due to irrelevance. The full texts of 226 articles were reviewed for eligibility, and 182 were excluded for the following reasons: non-healthy samples (n = 5), non-quantitative design (n = 14), lack of measures of overparenting or mental health (n = 92), absence of effect size data (n = 12), duplication publications (n = 21), and inaccessible full texts (n = 37). Ultimately, 44 articles met the inclusion criteria and were included in the final analysis.

### 2.2. Coding of the Studies

Each study was coded using a detailed coding scheme. All studies included in the meta-analysis were identified with an asterisk in the references section. The coding scheme was informed by prior meta-analyses on parenting and mental health (e.g., [75]) and refined through iterative discussion among the research team to ensure comprehensive coverage of relevant study characteristics.

Two trained research assistants independently coded all studies. Training involved a detailed review of the coding manual, practice coding of a subset of studies, and discussion to resolve ambiguities. Inter-rater reliability values > 0.75 across variables, indicating excellent agreement. Discrepancies were resolved through discussion with the first author and corresponding authors to reach consensus.

The specific coded information for each study included categorical variables such as country, offspring’s gender (male, female), parental gender (father, mother), informants of overparenting and mental health (offspring-report, parent-report), study design (cross-sectional or longitudinal), culture (non-Western, Western belonging to America, or Non-American Western), developmental stage (indexed by age; children, adolescents, or adults). It should be noted that parental gender was coded based on the information reported in the original studies, which predominantly distinguished between maternal and paternal roles. As such, non-binary, transgender, or other non-traditional parenting identities were not captured, reflecting the current limitations in the available literature.

### 2.3. Data Analysis

The *Pearson* correlation coefficient (r) was used to examine the relationship between overparenting and mental health variables such as depression, anxiety, life satisfaction, and subjective well-being. Utilizing the Comprehensive Meta-Analysis software version 3.3, *r* values from reported test statistics were calculated. Subsequently, the Pearson correlation coefficient r was converted to Fisher’s z correlations, ensuring a normal distribution of the sample for further analysis. Finally, we converted Fisher’s z correlations back to Pearson correlation coefficients r to represent our research results ([2]).

The data analysis comprised several key components, including heterogeneity testing of the meta-analysis, publication bias testing, main effect testing, moderating effect testing, and sensitivity analysis. Subgroup analysis was employed for categorical moderating variables. Given that our primary study was considered a random sample of the study population, a random-effects model was deemed appropriate for the meta-analysis ([20]). Publication bias can significantly impact the outcomes of meta-analysis; therefore, this study employed various methods to assess and address this bias. Funnel plots and Egger’s regression test were used to evaluate publication bias ([60]). In instances where studies exhibited a high risk of publication bias, the trim-and-fill method was employed for bias correction ([14]).

## 3. Results

### 3.1. Descriptive Statistics

A total of 44 studies were included in the current meta-analysis. Among these, 34 studies reported 64 effect sizes for the associations between overparenting and anxiety (*N* = 37,872; *M*_age_ = 17.61, *SD* = 3.56), with participants’ age ranging from 12.63 to 24.3 years. 32 studies provided 65 effect size for depression (*N* = 31,855; *M*_age_ = 20.98, *SD* = 2.97; age range = 12.63–24.88 years). 13 studies reported 36 effect size for life satisfaction (*N* = 17,293; *M*_age_ = 22.97, *SD* = 2.39; age range = 19.34–25 years), and three studies reported four effect size for subjective well-being (*N* = 1687; *M*_age_ = 18.15, *SD* = 2.71; age range = 16.59–21.28 years). Most studies were conducted in the United States (49.45%) and employed a cross-sectional design (83.33%). On average, 65% of participants were female. Approximately 79.44% did not distinguish between maternal and paternal overparenting. Regarding the age groups, most studies (85.56%) included a sample of emerging adults, and 14.44% of studies used a sample of adolescents. Table 1 presents a detailed summary of study characteristics.

### 3.2. Meta-Analysis of Overparenting and Indicators of Mental Health

The overall effect size indicating the association between overparenting and depression was found to be significant (*r* = 0.20, 95% CI [0.17, 0.24], *p* < 0.001), suggesting that a higher level of overparenting is linked with increased depression. Sensitivity analysis revealed that upon removal of any sample, the combined effect size fluctuated between 0.19 and 0.21, with no change in significance, indicating high stability in the estimated results of the meta-analysis. Examination of the funnel plot and the Egger test (*p* = 0.932) did not reveal any evidence of publication bias (see Table 2 and Appendix A). Moderator analysis results, as presented in Appendix A, indicate no significant moderation effects of variables except for the developmental stage (indexed by age), which yielded significant results (*k* = 65, *r* = 0.138, 95% CI [0.114, 0.163], *p* < 0.001). This finding indicates that the association between overparenting and depression varies across developmental stages, suggesting that the impact of overparenting may differ depending on whether the offspring is in childhood, adolescence, or adulthood.

Results showed that a significant positive but minor effect of overparenting and anxiety was found (*r* = 0.16, 95% CI [0.13, 0.19], *p* < 0.001), suggesting that a higher level of overparenting is associated with increased anxiety. Sensitivity analysis revealed high stability in the estimated results of the meta-analysis. Examination of the funnel plot (see Appendix A) and the Egger test (*p* = 0.003) indicated publication bias. The trim-and-fill method was employed to address publication bias, resulting in the inclusion of 11 virtual studies. The combined effect size remained statistically insignificant, and no reversal was observed, indicating the robustness of the combined effect size (*r* = 0.19, 95%CI [0.16, 0.23], *p* < 0.001). Moderator analyses showed that (see Appendix A) no significant moderation effects of all variables except for parental gender and developmental stage (indexed by age). Specifically, maternal overparenting showed a stronger link to anxiety than paternal or both. Age-group analysis revealed that the association was strongest in adults (*r* = 0.175, 95% CI [0.160, 0.189]), followed by adolescents’ overparenting (*r* = 0.113, 95% CI [0.099, 0.126]). These results suggest that the impact of overparenting on anxiety varies across developmental stages, highlighting the importance of considering offspring development rather than treating age merely as a categorical variable.

The results showed a significant but small effect of overparenting on life satisfaction (*r* = 0.09, 95% CI [0.038, 0.133], *p* < 0.001), with the high stability of the results found through sensitivity analysis and no publication bias found (Egger test *p* = 0.100) (see Appendix A). Moderator analysis showed that (see Appendix A) there are no significant effects of all variables. However, culture significantly moderated the relationship: non-Western cultural contexts showed a positive association (*r* = 0.15, 95% CI [0.123, 0.174]), while Non-American Western, cultural contexts and the American context showed nonsignificant or negative associations. Additionally, the moderation test for developmental stage (indexed by age) yielded significant results, with a trend toward stronger associations in older age groups, indicating that the impact of overparenting on life satisfaction may vary across childhood, adolescence, and adulthood.

The meta-analysis found no significant association between overparenting and subjective well-being (*r* = −0.01, 95% CI [−0.122, 0.097], *p* > 0.001), with stable results through sensitivity analysis despite publication bias (Egger test, *p* = 0.007) (also see Appendix A). The moderator analysis results, presented in Appendix A, revealed no significant effects for any of the variables.

## 4. Discussion

This meta-analysis systematically examined the association between overparenting and various mental health outcomes, including depression, anxiety, life satisfaction, and subjective well-being, based on studies published from 2003 to 2023. The findings suggest that overparenting is positively associated with both negative (depression, anxiety) and, to a lesser extent, positive (life satisfaction, subjective well-being) mental health indicators. Notably, age and culture emerged as significant moderators in these associations, offering new insights into the understanding of the relationship between overparenting and mental health, providing theoretical implications for parent–child relationships, and the intervention of mental health disorders.

### 4.1. Associations Between Overparenting and Mental Health

The current meta-analysis revealed a small but significant positive correlation between overparenting and various mental health indicators, including depression, anxiety, and life satisfaction. This finding aligns with prior studies conducted in East Asian contexts, such as South Korea, where overparenting is sometimes associated with favorable psychological outcomes ([35]; [69]). Despite its modest magnitude, this result contrasts with several U.S.-based studies that report detrimental effects of overparenting on mental health ([11]), highlighting the nuanced and potentially culturally contingent nature of overparenting. These differences underscore the need to interpret overparenting not solely as a risk factor, but as a contextually embedded parenting behavior that may have both adaptive and maladaptive consequences.

Importantly, cultural norms may shape how children perceive and evaluate parental behaviors ([34]). In collectivist cultures emphasizing filial piety and family interdependence ([35]), high parental involvement may be appraised as caring and supportive, which can buffer against negative psychological outcomes and even enhance life satisfaction ([76]). Conversely, in individualist cultures prioritizing autonomy and independence, similar behaviors may be perceived as intrusive or controlling, eliciting stress, frustration, or anxiety ([35]). These appraisal processes—shaped by culturally shared beliefs and values—likely mediate the relationship between overparenting and mental health outcomes, explaining why the same parenting behaviors can produce divergent effects across cultural contexts.

Consistent with prior research, overparenting was significantly and positively associated with depressive symptoms ([36]; [59]; [11]). From the perspective of *Self-Determination Theory* ([57]), this association may be attributed to the undermining of autonomy caused by excessive parental control, which can evoke feelings of helplessness and emotional distress ([15]; [50]). Similarly, overparenting was positively correlated with anxiety, reinforcing findings from earlier studies ([28]). According to *Separation-Individuation Theory* ([19]), excessive parental control can disrupt the normative process by which adolescents establish independence, thereby increasing their vulnerability to experiencing elevated stress and anxiety. These mechanisms help explain why overparenting exerts stronger negative effects as children age and seek greater autonomy. Interestingly, our findings contradict a recent meta-analysis that reported a negative association between overparenting and anxiety ([75]), suggesting that more research is needed to clarify this inconsistency.

The relationship between overparenting and life satisfaction was also positive, though the effect size was small. This finding diverges from several U.S.-based studies, which report that overparenting diminishes life satisfaction ([36]; [45]). However, it aligns with research conducted in collectivist cultures, where close parental involvement is more culturally accepted ([7]; [35]). In such settings, parental overinvolvement may be interpreted as warmth and support rather than intrusion, which can enhance psychological well-being. However, the association between overparenting and subjective well-being was found to be non-significant, which contrasts with prior research indicating a negative impact ([59]; [32]). This discrepancy may be due to limited sample size, measurement inconsistencies, or cross-cultural variation in the conceptualization of well-being. Future studies should explore these dimensions using more culturally sensitive and psychometrically robust tools.

In sum, the findings reveal a complex pattern in the relationship between overparenting and offspring mental health. While overparenting is associated with increased levels of depression and anxiety—possibly due to thwarted autonomy—it also appears to have a weak but positive association with life satisfaction, particularly in collectivist contexts. These mixed results highlight the importance of considering cultural context, developmental stage, and children’s subjective interpretations when evaluating the psychological impact of overparenting. Moreover, the non-significant findings regarding subjective well-being point to potential measurement issues or underexplored moderating variables. Though the lens of Dual-Factor Model (DFM), this study offers a more balanced and comprehensive understanding of mental health outcomes, laying the groundwork for future culturally informed intervention strategies.

### 4.2. Moderators of Associations Between Overparenting and Mental Health

The results of this meta-analysis indicate that multiple variables—including developmental stage (indexed by age), cultural context, parental gender, study design and offspring’s gender—significantly moderate the association between overparenting and offspring mental health. These findings expand upon previous research and challenge prior meta-analyses that reported no significant moderation effects, particularly with respect to developmental stage (indexed by age) and culture (e.g., [75]). The identification of these moderators highlights the need for more contextually informed approaches when examining the impact of overparenting. However, it should be noted that the majority of included studies (83.33%) employed cross-sectional designs. Therefore, the associations observed cannot be interpreted as causal, and caution is warranted when discussing potential causal mechanisms. Future research should utilize longitudinal or experimental designs to better examine the directionality and causality of the relationships between overparenting and mental health outcomes.

Developmental stage emerged as a significant moderator in the relationship between overparenting and both depression and anxiety. This aligns with SDT and SIT, suggesting that as children grow older, their increasing need for autonomy makes them more susceptible to the detrimental effects of intrusive parenting. Contrary to previous meta-analyses reporting no age-related moderation ([75]), our findings suggest that the psychological impact of overparenting becomes more pronounced with increasing age. This may reflect developmental shifts in autonomy needs: as children grow older, they increasingly seek independence ([67]), and intrusive parental behaviors may be more likely to generate psychological distress ([41]). These findings underscore the importance of implementing age-sensitive parenting strategies and interventions, particularly during adolescence and emerging adulthood, when autonomy is a developmentally salient aspect.

Parental gender also moderated the relationship between overparenting and psychological outcomes. Maternal overparenting was more strongly associated with offspring anxiety and depression than paternal overparenting, consistent with previous findings ([37]; [75]). One possible explanation is that mothers tend to be more involved in day-to-day caregiving, which may increase both the frequency and salience of overparenting behaviors ([47]). This heightened exposure could make children more sensitive to maternal overinvolvement, thereby amplifying its psychological effects.

Additionally, culture significantly moderates the association between overparenting and life satisfaction. Consistent with previous research, children raised in Eastern cultures, which emphasize filial piety, may perceive greater affection and warmth from their parents compared with those in Western cultures, which prioritize individualism ([35]). Mechanistically, cultural norms influence children’s appraisal of parental behaviors: supportive intent is emphasized in collectivist contexts, reducing the perception of intrusiveness, whereas autonomy-oriented norms in individualist contexts amplify the perception of parental overcontrol ([38]). This appraisal process likely mediates the relationship between overparenting and psychological outcomes, offering a theoretical explanation for the observed cross-cultural differences.

In summary, the present meta-analysis identifies several key moderators—developmental stage, culture, parental gender—that shape the relationship between overparenting and mental health outcomes. These moderators highlight the contextual variability of overparenting’s psychological consequences, emphasizing that its effects are not universally negative or positive, but instead depend on the developmental, cultural, and familial context. Understanding these moderators provides critical insights for tailoring parenting interventions and promoting culturally and developmentally appropriate mental health strategies. Moreover, the predominance of cross-sectional studies in the current literature underscores the importance of future longitudinal and experimental research to establish causality and clarify the temporal dynamics of overparenting’s impact on mental health.

Importantly, these findings have practical implications for parenting interventions. Specifically, interventions could aim to reduce developmentally inappropriate overparenting behaviors while maintaining supportive involvement. Age-sensitive strategies could promote autonomy in adolescents and emerging adults, whereas culturally tailored approaches could guide parents in balancing control and support according to collectivist or individualist norms. Additionally, given the stronger negative effects of maternal overparenting, parent-focused guidance could help optimize parental involvement to enhance offspring mental health outcomes.

### 4.3. Limitations and Future Directions

Despite its contributions, this study has limitations.

First, the limited number of studies on subjective well-being, coupled with insufficient reporting on demographic variables—including participants’ age and developmental stage—may constrain the generalizability and robustness of the findings. Notably, most existing studies focus on adolescents or emerging adults, while research on early childhood remains scarce ([69]). This gap limits our understanding of how overparenting affects mental health across the full developmental stages. Future research should include younger children and provide detailed demographic information to allow for more comprehensive moderation analyses.

Second, the restricted sample characteristics and demographic reporting may weaken the precision of moderator analyses. Without detailed information on participants’ age, socioeconomic background, or other contextual factors, it is challenging to determine how these variables interact with overparenting to influence mental health outcomes. Addressing these gaps in future studies will enhance the interpretability and applicability of meta-analytic findings.

Third, the majority of included studies relied on adult children’s self-reports of parental overparenting, which may introduce measurement bias. For example, individuals with higher levels of depression or anxiety might perceive or recall their parents’ behaviors more negatively, potentially inflating observed associations. Future research should employ multi-informant, observational, or longitudinal designs to better establish causality and reduce such bias.

Fourth, as noted in Methods, parental gender was coded as maternal or paternal based on the information reported in the original studies, and non-binary, transgender, or other non-traditional parenting identities were not captured. Future research should consider more inclusive family structures to advance theoretical understanding and enhance the generalizability of findings.

Fifth, most included studies (83.33%) employed cross-sectional designs, which precludes causal inferences. Future research should employ longitudinal or experimental designs to clarify the directionality and causality of associations between overparenting and mental health outcomes.

Finally, although cultural differences in parenting norms and their implications for mental health were considered, cultural context was operationalized only as a broad East–West distinction (collectivist vs. individualist). This coarse classification may overlook more nuanced family-level, regional, or subcultural differences that influence parenting behaviors and offspring mental health outcomes. Future research should adopt more granular and multidimensional measures of culture to better capture its impact and advance both theoretical understanding and intervention strategies.

## 5. Conclusions

To our knowledge, the current study is the first systematic examination of overparenting and its positive and negative indicators of mental health. The findings revealed that overparenting exerts a double-edged effect on mental health, which varies across offspring’s developmental stages (indexed by age), parental gender, and cultural background. This study broadens the scope of mental health indicators and provides a comprehensive understanding of the associations between overparenting and mental health. Moreover, the findings offer important insights for the design and implementation of developmentally and culturally sensitive parenting interventions aimed at promoting mental health.

## Figures and Tables

**Figure 1 behavsci-15-01235-f001:**
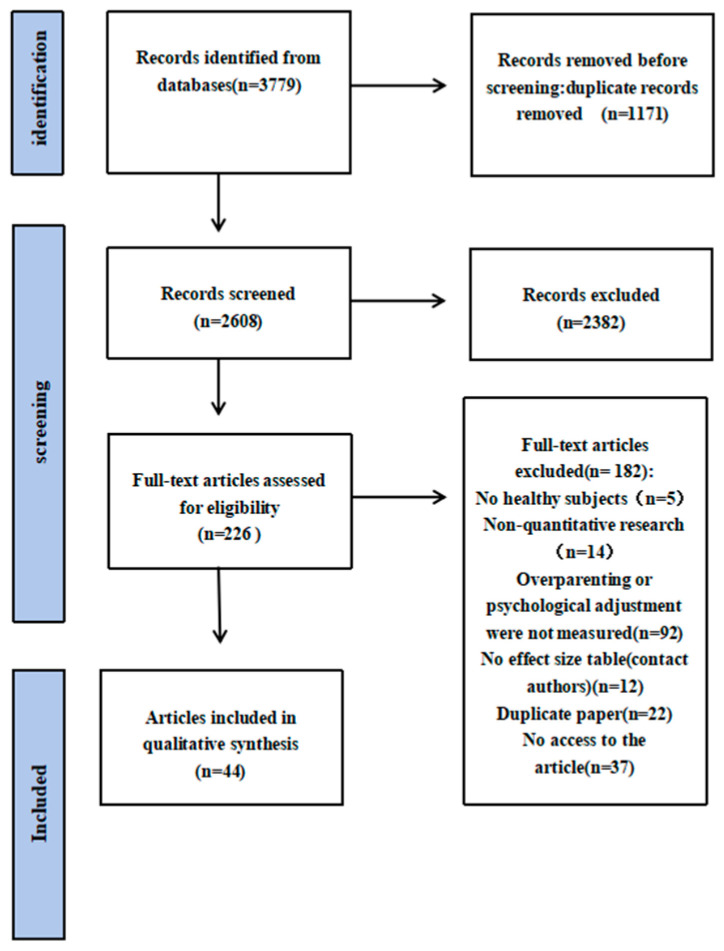
The PRISMA flow chart was used to systematically review the relationship between overparenting and mental health.

**Table 1 behavsci-15-01235-t001:** Demographic characteristics of studies (are labelled with *Asterisks in the References).

First Author and Year	N	Country	Cultures	Design	Parental Gender	Percent of Female	Age Group	Average Age of Offspring	Informants
[1] ([1])	71	Maldives	NW	C	Both	70%	adult	N/A	offspring
[3] ([3])	187	Portugal	N-A W	C	Both	64.70%	adult	21.20	offspring
[5] ([5])	76	USA	W	L	Mother	48.68%	adult	25	offspring
[6] ([6])	623	China	NW	C	Both	45.75%	adolescent	16.04	Both
[8] ([8])	637	USA	W	C	Both	67%	adult	20.03	offspring
[11] ([11])	294	USA	W	C	Both	81.97%	adult	20.54	offspring
[17] ([17])	194	USA	W	C	Both	64.95%	adult	N/A	offspring
[18] ([18])	418	China	NW	L	Both	80.1%	adult	18.71	offspring
[4] ([4])	173	Portugal	N-A W	C	Both	69.40%	adult	23.08	offspring
[21] ([21])	305	Korea	NW	C	Both	100%	adult	21.94	offspring
[23] ([23])	432	USA	W	C	Both	89.60%	adult	20.21	offspring
[24] ([24])	414; 612	USA; China	W; NW	C	Both	92%; 69%	adult	20.38; 20.21	offspring
[25] ([25])	460	Canada	N-A W	C	Both	43.91%	adult	18.33	offspring
[27] ([27])	213 pairs	USA	W	C	Both	65.80%	adult	20.63	Both
[26] ([26])	412	USA	W	L	Both	60.40%	adult	24.3	Both
[28] ([28])	442	USA	W	C	Both	68.10%	adult	20.28	offspring
[30] ([30])	215; 171	USA; Korea	W; NW	C	Mother	N/A	adult	19.61; 21.95	offspring
[32] ([32])	118	USA	W	C	Both	83.10%	adult	19.82	offspring
[33] ([33])	412	USA	W	C	Both	50.70%	adult	21.28	offspring
[35] ([35])	562	Korea	NW	C	Both	47.86%	adult	24.88	offspring
[37] ([37])	1735	China	NW	C	Both	47.40%	child	12.63	offspring
[40] ([40])	1074	China	NW	L	Both	46.80%	adolescent	12.66	offspring
[42] ([42])	473	USA	W	C	Both	N/A	adult	19.78	offspring
[43] ([43])	539	USA	W	C	Both	92.50%	adult	20.18	offspring
[44] ([44])	302	USA	W	C	Both	64.90%	adult	21.57	offspring
[46] ([46])	458	USA	W	L	Both	51%	adult	19	offspring
[48] ([48])	87	USA	W	C	Mother	63%	adult	N/A	offspring
[49] ([49])	313	USA	W	C	Both	82.40%	adult	19.55	Both
[50] ([50])	360	USA	W	C	Both	83.60%	adult	19.93	offspring
[51] ([51])	602	Italian	N-A W	C	Both	59%	adolescent	16.59	offspring
[52] ([52])	158	USA	W	C	Mother	74.7%	adult	20.28	offspring
[53] ([53])	461	USA	W	C	Mother	80.80%	adult	19.66	offspring
[54] ([54])	282	USA	W	C	Both	71%	adult	19.87	offspring
[59] ([59])	297	USA	W	C	Mother	88%	adult	19.34	offspring
[58] ([58])	446	USA	W	C	Both	73.10%	adult	19.59	offspring
[63] ([63])	653 pairs	USA	W	C	Both	81%	adult	20.03	Both
[61] ([61])	282; 281	USA; China	W; NW	C	Both	62.1%; 71.9%	adult	20.67; 19.83	offspring
[64] ([64])	402	Turkey	NW	C	Both	82.10%	adult	21.31	offspring
[66] ([66])	120	USA	W	C	Both	59.20%	adult	N/A	offspring
[71] ([71])	648	China	NW	C	Both	50.30%	adult	21	offspring
[72] ([72])	2041	China	NW	C	Both	52.60%	adolescent	14.11 ± 2.42	offspring
[73] ([73])	104	USA	W	C	Mother	77.88%	adult	19.15	offspring
[77] ([77])	156	USA	W	C	Both	71.15%	adult	N/A	offspring

*Note.* C = cross-sectional; L = longitudinal. N/A = not available; W = Western belonging to America; NW = non-Western; N-A W = Non-American Western.

**Table 2 behavsci-15-01235-t002:** Results of heterogeneity and publication bias.

Relationships	k	N	r	Heterogeneity Test			Publication Bias Test		
				Q	df	*I* ^2^	Egger’s Intercept	SE	95%CI
depression	65	31,855	0.20	768.78 **	64	91.67%	−0.12	1.42	[−2.97, 2.72]
anxiety	34	37,872	0.16	719.05 **	63	91.24%	3.80	1.24	[1.31, 6.28]
life satisfaction	13	17,293	0.09	315.83 **	35	90.05%	−5.03	2.98	[−11.08, 1.02]
subjective well-being	3	1687	−0.01	13.36	3	77.54%	−5.59	0.49	[−7.68, −3.49]

*Note*: ** *p* < 0.01.

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
