# Peer review of "Associations Between Overparenting and Offspring’s Mental Health: A Meta-Analysis of Multiple Moderators"

_behavsci, 2025, doi:10.3390/bs15091235_

Round 1
Reviewer 1 Report
Comments and Suggestions for Authors
This is a meta-analysis examining the links between overparenting and children’s mental health.
There are several ways in which this paper could be improved.
- First, the introduction strays from its organization, introducing different literature documenting the mixed findings between overparenting and outcomes in each section, regardless of their stated focus.
- Second, it is critically important that overparenting be defined more operationally. The label implies “more than” and in that case, it is a relative label. The paper would benefit from a thorough discussion of what overparenting is and is not. This is lacking.
- Mediator and moderator are sometimes misused. If child/offspring age is a moderator, it is development that is the issue (i.e., age is not categorical but is a marker of a developmental process).
- What is the rationale for the articles to specifically require all participants to be healthy? What does that mean? A literal interpretation would mean that parents and children could not suffer from any chronic (e,g., asthma) or acute (e.g., a cold) physical or mental health conditions; but could have chronic disability as long as it was not a sign of a health condition (e.g., intellectual disability, deafness). It is a very odd inclusion criterion that I have never seen before. If the authors actually mean something more restrictive, please change the language. If the language is correct, please specify the nature of the samples that were excluded by requiring this.
- Given that the vast majority of studies included collected data only from the adult children who reported on their parenting, it is critical to address the possibility of measurement bias and how that could lead to spurious associations (such that depressed adults are tend view their parents more negatively).
Author Response
Comment 1: First, the introduction strays from its organization, introducing different literature documenting the mixed findings between overparenting and outcomes in each section, regardless of their stated focus.
Response 1: Thank you for pointing this out. I agree with this comment. We carefully revised the Introduction to improve its organization and avoid redundancy. Specifically, we restructured the section:
Firstly, we revised the “Empirical studies have documented both negative and positive associations between overparenting and offspring’s mental health, mainly focused on the negative aspect of mental health, but the findings have been inconsistent.” into “Empirical studies have primarily focused on the negative associations between overparenting and offspring’s mental health, yet some research has also identified posi-tive links, resulting in inconsistent findings.”. Also, we added the “These mixed results highlight the need for a more comprehensive framework to capture both the detrimental and beneficial aspects of mental health.” after this sentence. Please see Lines 39-43.
Secondly, we revised the section 1.2, we separated evidence of negative outcomes and positive or neutral outcomes into distinct paragraphs. In addition, the final paragraph of 1.2 now highlights the inconsistency of findings and introduces the need to examine potential moderators, creating a natural transition to section 1.3. Please see Lines 88-90, Lines 99-101, Lines 110-115.
Thirdly, we deleted the first paragraph in the section 1.3, the following sentence” As highlighted in the previous section, the associations between overparenting and offspring’s mental health appear to vary across studies and contexts. To better un-derstand this variability, it is essential to examine potential moderating factors that may influence when and why overparenting leads to detrimental, neutral, or even beneficial outcomes.” replaced the original sentences. Please see Lines 117-120.
Overall, these changes address the reviewer’s concern by providing clearer focus, logical flow, and structured presentation of mixed findings, while avoiding repetition between the general introduction and 1.2. Please see Lines 78-87.
Comment 2: Second, it is critically important that overparenting be defined more operationally. The label implies “more than” and in that case, it is a relative label. The paper would benefit from a thorough discussion of what overparenting is and is not. This is lacking.
Response 2:Thanks for your comments, we thank the reviewer for highlighting the need for a more operational definition of overparenting. In response, we have added a new subsection “1.1 Conceptions of Overparenting” in the Introduction, providing a detailed, operationalized description of overparenting. Specifically, we:
- Clarified the relative nature of overparenting, emphasizing that it refers to parental behaviors exceeding age-appropriate or culturally normative expectations.
- Specified key behavioral dimensions, including excessive control, involvement, protection, anticipatory problem-solving, risk aversion, and unnecessary assistance.
- Distinguished overparenting from normative parental involvement, providing concrete examples of what constitutes typical parenting versus overparenting.
These revisions ensure that overparenting is clearly defined both conceptually and operationally, directly addressing the reviewer’s concern. Please see Lines 56-76.
Comment 3: Mediator and moderator are sometimes misused. If child/offspring age is a moderator, it is development that is the issue (i.e., age is not categorical but is a marker of a developmental process).
Response 3: Thank you kindly pointed that “mediator and moderator are sometimes misused”, we revised all “mediator” into “moderator” correctly. Please see the revised manuscript.
In addition, we thank the reviewer for pointing out this important issue regarding the use of age as a moderator. We agree that age should not be treated merely as a categorical grouping variable but rather as an indicator of developmental stage. Accordingly, we have revised the manuscript as follows.
In the Introduction (Lines 143, 156, 158): We replaced “age group” with “developmental stage” and emphasized that children’s age reflects developmental processes that influence how overparenting is experienced (see revised text: “Consequently, it remains unclear whether developmental stage moderates the association between overparenting and mental health…”). This modification underscores that our interest lies in developmental differences rather than simple age categories.
In the Coding section (Line 228): We clarified that studies were coded into three broad developmental stages—childhood, adolescence, and adulthood—rather than only by age groups. This coding approach was adopted to better capture developmental variation in the associations between overparenting and mental health.
In the Results、Discussion and Conclusion section (Lines 280-284; Line 295-296; Line 299-301; Line 309-312; Line 388-389; Line 393; Line 401; Line 431- 432; Line 491): All moderator analyses now refer to developmental stage, with interpretations emphasizing developmental differences in overparenting’s impact.
Comment 4: What is the rationale for the articles to specifically require all participants to be healthy? What does that mean? A literal interpretation would mean that parents and children could not suffer from any chronic (e,g., asthma) or acute (e.g., a cold) physical or mental health conditions; but could have chronic disability as long as it was not a sign of a health condition (e.g., intellectual disability, deafness). It is a very odd inclusion criterion that I have never seen before. If the authors actually mean something more restrictive, please change the language. If the language is correct, please specify the nature of the samples that were excluded by requiring this.
Response 4: We appreciate the reviewer’s careful reading and insightful comment. We agree that the original phrasing“All participants were required to be healthy”was ambiguous and could be misinterpreted. To clarify, we revised the inclusion criterion to specify that participants were included if they did not have clinically diagnosed psychiatric or neurological disorders, while minor or transient physical conditions (e.g., common colds) were not grounds for exclusion. The rationale for this criterion is to avoid confounding the associations between overparenting and mental health outcomes, ensuring that observed effects reflect typical developmental processes rather than pre-existing clinical conditions. The Methods section has been updated accordingly, Please see revised Inclusion Criteria, Item d, Line 196-199.
Comment 5: Given that the vast majority of studies included collected data only from the adult children who reported on their parenting, it is critical to address the possibility of measurement bias and how that could lead to spurious associations (such that depressed adults are tend view their parents more negatively).
Response 5: We appreciate the reviewer’s insightful comment. We agree that the reliance on adult children’s self-reports may introduce measurement bias, as individuals with higher levels of depression or anxiety could perceive or recall their parents’ behaviors more negatively, potentially inflating the observed associations. To address this, we have added a discussion in the Limitations section, stating:
“Third, majority of included studies relied on adult children’s self-reports of parental overparenting, which may introduce measurement bias. For example, individuals with higher levels of depression or anxiety might perceive or recall their parents’ behaviors more negatively, potentially inflating observed associations. Future research should employ multi-informant, observational, or longitudinal designs to better establish causality and reduce such bias..”Please see Lines 465-470.

Reviewer 2 Report
Comments and Suggestions for Authors
This manuscript presents a comprehensive meta-analysis examining the association between overparenting and various indicators of offspring mental health. Here are a few comments:
- The title is somewhat abstract and not sufficiently informative. I would prefer a more straightforward and descriptive title.
- The definitions of key constructs like “overparenting” and “positive mental health” could be better clarified in the Introduction.
- While the study examines parental and offspring gender as moderators, terms such as “maternal” and “paternal” are used exclusively. Please consider a brief note acknowledging the potential exclusion of non-binary or non-traditional parenting identities in current literature.
- As the majority of included studies are cross-sectional (83.33%), caution should be emphasized more clearly when interpreting causal relationships, and a call for longitudinal research should be add in the discussion.
-
The discussion could be enriched by elaborating on the mechanisms through which cultural norms shape the perception and impact of overparenting.
- Although the manuscript outlines several theoretical models in the introduction—including Self-Determination Theory, Separation-Individuation Theory, and the Dual-Factor Model of Mental Health, these frameworks are only lightly referenced in the discussion.I encourage the authors to more actively engage with the theoretical frameworks introduced earlier in the paper in the discussion.
Author Response
Comment 1: The title is somewhat abstract and not sufficiently informative. I would prefer a more straightforward and descriptive title.
Response 1: Thanks for your comments, we agree with them. We have revised the title to make it more informative and precise. The new title“Associations between Overparenting and Offspring’s Mental Health: A Meta-Analysis of Multiple Moderators” clearly indicates the study topic, methodology, and scope of analysis. It reflects that the meta-analysis examines the relationship between overparenting and offspring mental health, considering multiple moderating factors, including developmental stage, gender, cultural background, and study design. Please see the title in the revised manuscript.
Comment 2: The definitions of key constructs like “overparenting” and “positive mental health” could be better clarified in the Introduction.
Response 2: We thank the reviewer for this valuable suggestion. In the revised manuscript, firstly, we added the 1.1 section to clarify the key constructs of “overparenting”, please see the revised manuscript, Lines 55-76.
Secondly, we have clarified the definition and operationalization of positive mental health in the Introduction (Section 1.2). Specifically, we now define positive mental health as “the scientific study of those positive strengths and virtues that enable people and communities to reach optimal levels of health, happiness, and well-being” (Seligman & Csikszentmihalyi, 2000; Keyes, 2005). To link this concept with our meta-analytic measures, we explicitly state that life satisfaction and subjective well-being serve as positive indicators and provide measurable manifestations of positive mental health. We also clarified that mental health in our study encompasses both negative indicators (e.g., anxiety and depressive symptoms) and positive indicators, allowing us to capture the full spectrum of overparenting’s impact and examine potential “double-edged” effects. These changes aim to make the constructs conceptually and operationally clear. Please see Lines 79-87.
Comment 3: While the study examines parental and offspring gender as moderators, terms such as “maternal” and “paternal” are used exclusively. Please consider a brief note acknowledging the potential exclusion of non-binary or non-traditional parenting identities in current literature.
Response 3: We appreciate the reviewer’s insightful comment. In response, we have added clarifications in both the Methods and Limitations sections.
In the Methods section, we added a sentence “It should be noted that parental gender was coded based on the information reported in the original studies, which predominantly distinguished between maternal and paternal roles. As such, non-binary, transgender, or other non-traditional parenting identities were not captured, reflecting the current limitations in the available literature.”. Please see Lines 229-233.
In the Limitations section, we acknowledged this restriction and suggested that future research should consider more inclusive family structures to advance theoretical understanding and enhance the generalizability of findings. We added “Fourth, as noted in Methods, parental gender was coded as maternal or paternal based on the information reported in the original studies, and non-binary, transgender, or other non-traditional parenting identities were not captured. Future research should consider more inclusive family structures to advance theoretical understanding and enhance the generalizability of findings.”. Please see Lines 471-475.
Comment 4: As the majority of included studies are cross-sectional (83.33%), caution should be emphasized more clearly when interpreting causal relationships, and a call for longitudinal research should be added in the discussion.
Response 4: We thank the reviewer for this important comment. In response, we have clarified in both the Discussion and Limitations sections that majority of included studies (83.33%) employed cross-sectional designs. We explicitly note that, due to this, causal inferences cannot be drawn from the observed associations. We also added a recommendation that future research should employ longitudinal or experimental designs to clarify the directionality and causality of the relationships between overparenting and mental health outcomes. Please see Lines 395-400; Lines 476-479.
Comment 5: The discussion could be enriched by elaborating on the mechanisms through which cultural norms shape the perception and impact of overparenting.
Response 5: We appreciate the reviewer’s insightful comment regarding the role of cultural norms. In response, we have expanded the Discussion section (see Section 4.1 and 4.2) to elaborate on potential mechanisms. Specifically, we highlight that in collectivist cultures, such as East Asian contexts, overparenting behaviors may be interpreted as expressions of parental warmth, support, and familial interdependence, rather than as intrusive or controlling behaviors. This interpretation may buffer the negative impact of overparenting on offspring mental health and, in some cases, even contribute to positive outcomes, such as higher life satisfaction. In contrast, in individualistic Western cultures, where autonomy and independence are highly valued, similar parental behaviors may be perceived as overcontrolling, leading to elevated anxiety and depressive symptoms. These cultural differences underscore the importance of considering culturally embedded perceptions and values when evaluating the psychological impact of overparenting. The added discussion clarifies how cultural norms may shape both the perception and consequences of overparenting, providing a more nuanced understanding of its context-dependent effects. Please see Lines 339-348; Lines 425-430.
Comment 6: Although the manuscript outlines several theoretical models in the introduction—including Self-Determination Theory, Separation-Individuation Theory, and the Dual-Factor Model of Mental Health, these frameworks are only lightly referenced in the discussion. I encourage the authors to more actively engage with the theoretical frameworks introduced earlier in the paper in the discussion.
Response 6: Thanks for the reviewer’s careful comments, we appreciate the reviewer’s insightful comment regarding the integration of theoretical frameworks. In the revised manuscript, we have more actively engaged with the theoretical models introduced in the Introduction throughout the Discussion. Specifically, we clarified how Self-Determination Theory explains the negative impact of overparenting on autonomy and competence, how Separation-Individuation Theory accounts for developmental vulnerability to parental intrusiveness. These revisions provide a more mechanistic understanding of how overparenting influences offspring mental health and link the empirical findings directly to the theoretical frameworks. Please see Line 351; Lines 355-360; Line 384.
Reviewer 3 Report
Comments and Suggestions for Authors
Thank you for the opportunity to review this manuscript. The study addresses an interesting and meaningful topic in child development, particularly the role of overparenting in mental health. The manuscript has multiple strengths, including being well-written and inclusive of a wide range of studies for the meta-analysis. Despite this, there are multiple issues with the manuscript in the current form, and I have provided by section specific thoughts about this below.
Introduction:
The overparenting definition with same sentence structure and word choice is repeated in the second paragraph and 1.1 first paragraph. Further, the term overparenting itself feels problematic. It presumes a negative parenting stance while, at the same time, the authors are saying there are potential factors (ie – culture) that can result in positive outcomes. Thus, more care and thought needs to be given to this construct and the emerging – newer and more nuanced – view of what the specific parenting behaviors being examined are and what whether or not they are problematic.
How culture is presented and explored here is insufficient. Multiple studies find that the same parenting practices that are helpful in some communities are harmful in others. The statement about cultural differences being individualistic versus collectivist (East v West) is too broad and brief. Multiple studies find that the family – as opposed to the prevailing society – play an outsized role in shaping parenting impacts. For example, regional differences, immigrant populations, etc have different ways of viewing cultural background. Thus, cultural context interacts with families within them to produce the parenting practices used and whether or not they adaptively support development. Much more nuance is needed in exploring this construct.
Some rationale needs to be provided as to why developmentally there might (or might not) be age effects. For example, children’s need for guidance in early childhood is quite different then the autonomy needs of adolescence. More theoretical justification needed for this. Additionally, the current study notes that study design will also be a moderator of focus but no rationale for this is presented.
Method/results:
I will trust the methodologists who are also reviewing this to weigh in on whether the analyses are conducted and interpreted correctly. There are a couple other issues in this section that I noticed that need attention.
More information about the coders and coding process is needed. How was the coding scheme developed? How many coders coded the studies? How were coders trained, and what is the reliability between the coders?
Please check the grammar and punctuation (e.g. line 235 [delete “was found”], 245[suggest deleting “or both”], 256 [I suggest using Non-American Western cultural contexts throughout the paper instead of Western, but not American, cultural contexts]).
Discussion:
There is a limited expansion on the findings and what they mean, particularly as they relate to the culture and developmental period results. This is due in part that the rationale for the role of those constructs was not adequately theorized in the introduction. Doing that in the set up, then returning to them here, would better contextualize the results.
Several times throughout the manuscript, the authors stated that these findings would inform parenting interventions. This point, however, was not made clear and needs to be expanded on if this is the case.
A future direction mentioned earlier in the paper but not returned to is the lack of research on overparenting and early child mental health. This should be expanded on here.
Author Response
Comment 1: Introduction: The overparenting definition with same sentence structure and word choice is repeated in the second paragraph and 1.1 first paragraph. Further, the term overparenting itself feels problematic. It presumes a negative parenting stance while, at the same time, the authors are saying there are potential factors (ie – culture) that can result in positive outcomes. Thus, more care and thought needs to be given to this construct and the emerging – newer and more nuanced – view of what the specific parenting behaviors being examined are and what whether or not they are problematic.
Response 1: We appreciate the reviewer’s valuable feedback on conceptual clarity. In the revision, firstly, we deleted the second paragraph and original 1.1 first paragraph. Secondly, we added a paragraph 1.1 section titled “Conceptions of Overparenting”, which describe the construct of overparenting, compare the overparenting and other parenting behaviors. Importantly, we note that the psychological impact of overparenting may vary by developmental stage and cultural context, acknowledging that in some circumstances, such behaviors may be perceived as supportive rather than intrusive. Please see Lines 55-76.
Comment 2: How culture is presented and explored here is insufficient. Multiple studies find that the same parenting practices that are helpful in some communities are harmful in others. The statement about cultural differences being individualistic versus collectivist (East v West) is too broad and brief. Multiple studies find that the family – as opposed to the prevailing society – play an outsized role in shaping parenting impacts. For example, regional differences, immigrant populations, etc, have different ways of viewing cultural background. Thus, cultural context interacts with families within them to produce the parenting practices used and whether or not they adaptively support development. Much more nuance is needed in exploring this construct.
Response 2: Thanks for your essential and valuable suggestion. We sincerely appreciate the reviewer’s careful and valuable comment regarding the complexity of cultural context. We agree that culture operates at multiple levels—including societal, regional, and family—which may differentially influence parenting practices and their impacts. In our meta-analysis, we followed the approach commonly adopted in prior published studies (e.g., Zhang &Ji, 2023) by operationalizing culture as a broad East–West distinction, representing collectivist versus individualist contexts. We acknowledge that this approach does not capture finer-grained cultural variations, such as family-level or regional differences. Accordingly, we have added a statement to the Limitations section to highlight this point and recommend that future research consider more nuanced and multidimensional measures of culture to better understand its moderating role in overparenting and offspring mental health outcomes. Please see Lines 480-486.
Comment 3: Some rationale needs to be provided as to why developmentally there might (or might not) be age effects. For example, children’s need for guidance in early childhood is quite different then the autonomy needs of adolescence. More theoretical justification needed for this. Additionally, the current study notes that study design will also be a moderator of focus but no rationale for this is presented.
Response 3: Thanks for your valuable suggestion. Correspondingly, we have revised Section 1.3 to provide a theoretical rationale: in early childhood, parental guidance is generally supportive, whereas adolescents and emerging adults increasingly seek autonomy, making overparenting potentially intrusive and harmful (Ryan & Deci, 2000; Grotevant & Cooper, 1986). We also clarified that study design is considered a moderator to account for potential methodological differences across studies. These revisions address the reviewer’s concern regarding theoretical justification for developmental stage and study design as moderators. Please see Lines 145-152.
Comment 4: I will trust the methodologists who are also reviewing this to weigh in on whether the analyses are conducted and interpreted correctly. There are a couple other issues in this section that I noticed that need attention. More information about the coders and coding process is needed. How was the coding scheme developed? How many coders coded the studies? How were coders trained, and what is the reliability between the coders?
Response 4: Thanks for your seriously comments, we agreed it completely. Accordingly, we followed some prior studies, and added the following sentences in the 2.2 Coding of the Studies section in the revised manuscript.
“The coding scheme was informed by prior meta-analyses on parenting and mental health (e.g., Zhang & Ji, 2023) and refined through iterative discussion among the re-search team to ensure comprehensive coverage of relevant study characteristics.
Two trained research assistants independently coded all studies. Training involved a detailed review of the coding manual, practice coding of a subset of studies, and dis-cussion to resolve ambiguities. Inter-rater reliability values >.75 across variables, indi-cating excellent agreement. Discrepancies were resolved through discussion with the first author and corresponding authors to reach consensus.” Please see Lines 215-223.
Comment 5: Please check the grammar and punctuation (e.g. line 235 [delete “was found”], 245[suggest deleting “or both”], 256 [I suggest using Non-American Western cultural contexts throughout the paper instead of Western, but not American, cultural contexts]).
Response 5: Thanks for your cautious suggestions, we revised those in the revised manuscript. Please see the revised manuscript.
Comment 6: There is a limited expansion on the findings and what they mean, particularly as they relate to the culture and developmental period results. This is due in part that the rationale for the role of those constructs was not adequately theorized in the introduction. Doing that in the set up, then returning to them here, would better contextualize the results.
Response 6: Thanks for your comments, we all agreed it.
Referring to the cultures in the associations between overparenting and mental health in the Introduction section, we thank the reviewer for highlighting the need for a more nuanced discussion of cultural mechanisms. We agree that cultural context is important in shaping the perception and impact of overparenting. In response, we have elaborated on the mechanisms linking culture to mental health outcomes in the Discussion (see Section 4.2). We believe this addition adequately addresses the reviewer’s concern. We opted not to further expand on cultural mechanisms in the Introduction to avoid repetition and to maintain a clear and concise presentation of theoretical frameworks. Please see Lines 339-348; Lines 425-430.
Referring to the Developmental stage, similar with comments 3, we revised in the manuscript, please see Lines 145-152.
Comment 7: Several times throughout the manuscript, the authors stated that these findings would inform parenting interventions. This point, however, was not made clear and needs to be expanded on if this is the case.
Response 7: Thanks for the scientific comments, we couldn’t agree it any more. We agree that clarifying the practical implications of our findings is critical. In response, we have added a dedicated paragraph in the Discussion section (see the last paragraph) that explicitly outlines how our results can inform culturally and developmentally sensitive parenting interventions. Specifically, we discuss age-sensitive strategies, culturally tailored approaches, and guidance for optimizing parental involvement to promote offspring mental health.
“Importantly, these findings have practical implications for parenting interventions. Specifically, interventions could aim to reduce developmentally inappropriate overparenting behaviors while maintaining supportive involvement. Age-sensitive strategies could promote autonomy in adolescents and emerging adults, whereas cul-turally tailored approaches could guide parents in balancing control and support ac-cording to collectivist or individualist norms. Additionally, given the stronger negative effects of maternal overparenting, parent-focused guidance could help optimize pa-rental involvement to enhance offspring mental health outcomes.”. Please see Lines 441-448.
Comment 8: A future direction mentioned earlier in the paper but not returned to is the lack of research on overparenting and early child mental health. This should be expanded on here.
Response 8: Thank you for nice comments, we agree the point you mentioned. As a result, we reconstructed and revised the Limitation section, please see Lines 451-458.
Round 2
Reviewer 3 Report
Comments and Suggestions for Authors
I commend the authors for doing a very thorough and thoughtful revision. My prior comments have all been sufficiently addressed. The manuscript is much improved and will make a positive contribution to the field.